# A Simple Method for the Production of Human Skin Equivalent in 3D, Multi-Cell Culture

**DOI:** 10.3390/ijms21134644

**Published:** 2020-06-30

**Authors:** Łukasz Szymański, Krystyna Jęderka, Aleksandra Cios, Martyna Ciepelak, Aneta Lewicka, Wanda Stankiewicz, Sławomir Lewicki

**Affiliations:** 1Department of Molecular Biology, Institute of Genetics and Animal Biotechnology, Polish Academy of Science, Postępu 36A, 05-552 Magdalenka, Poland; lukas91.sz@gmail.com; 2Department of Regenerative Medicine and Cell Biology, Military Institute of Hygiene and Epidemiology Kozielska 4, 01-163 Warsaw, Poland; krystyna.jederka@wihe.pl; 3Department of Microwave Safety, Military Institute of Hygiene and Epidemiology, Kozielska 4, 01-163 Warsaw, Poland; aleksandra.cios@wihe.pl (A.C.); martyna.ciepielak@wihe.pl (M.C.); wanda.stankiewicz@wihe.pl (W.S.); 4Laboratory of Food and Nutrition Hygiene, Military Institute of Hygiene and Epidemiology, Kozielska 4, 01-163 Warsaw, Poland; aneta.lewicka@wihe.pl

**Keywords:** 3D model, skin equivalent, commercially available cell lines, differentiated keratinocytes, cytokines

## Abstract

An important problem for researchers working in the field of dermatology is the preparation of the human skin equivalent (HSE). Here, we describe a simple and reliable protocol for preparing a skin model from the commercially available cell lines: keratinocytes, fibroblasts, and melanocytes. Importantly, in our 3D model, the keratinocytes are diverse that brings this model closer to the natural skin. For the production of HSE, we used available primary PCS-200-010, PCS-201-010, PCS-200-013, and immortalized CRL-4048 and CRL-4001 cell lines. We used genipin, which is necessary for collagen cross-linking and studied its cytotoxicity for keratinocytes and fibroblasts. The addition of 20 μM genipin reduced the shrinkage of the collagen in the constructs from 59% to 24% on day 12 of the culture of the construct. A higher concentration (80–200 µM) of genipin reduced shrinkage by 14% on average. Genipin in concentration 10 μM and below was not cytotoxic to the keratinocytes, and 150 μM and below to the fibroblasts. Hematoxylin and eosin staining showed that the morphology of HSEs was identical to that of native human skin. The immunohistochemical staining of the constructs showed the presence of vimentin-positive fibroblasts in the skin layer, while the melanocytes were in the epidermis and in the basal layer. We observed that the longer differentiation of constructs led to the higher secretion of GM-CSF, IL-10, IL-15, IL-1α, IL-6, IL-7, IL-8, and MCP-1. We also observed that the longer time of differentiation led to a more stable secretion of all analytes, which was reflected in the coefficient of variation. We described here a simple, reliable, and cost-effective production of the full-thickness human skin equivalents that can be used in the research and industry. With the global trend to decrease animal use for the research and testing, our HSE could be a useful testing tool and an alternative research model.

## 1. Introduction

One of the biggest challenges for the researchers working in the field of dermatology is the construction of a model of human skin. Until now, most skin models are prepared from human skin biopsies, which may be quite problematic because of the individual differences between the donors. The first noticeable difference between the individuals is skin pigmentation [1] and its elasticity, not to mention structural differences between old and young skin [2]. The main problem of recreating the human skin is the inclusion of anciliary components such as dermal glands, and hair follicles, although simple full-thickness skin models having both dermal and epidermal components are available [3].

The first studies on the development of the epidermis met with deep skepticism among clinical practitioners [4]. Some of the studies described only ex vivo reconstructions of the epidermis with keratinocytes and melanocytes [5,6,7]. Because the two-dimensional monocultures don’t give satisfying results [8], the three-dimensional (3D) skin models are becoming more desirable for future therapeutic applications, for example in skin repair or tumor models, and also for the testing of medications and cosmetics.

Moreover, different cell types cultured separately behave differently than those in the co-culture. In the recently developed 3-D models of skin, the fibroblasts are “embedded” in the collagen matrix forming the “dermis”, while the “epidermis” consists of a layer of differentiated keratinocytes. The epidermis is separated from the dermis by a functional basement membrane. In these 3-D models, similar to the human skin obtained from patients, the melanocytes are in the basal layer [9]. The location of melanocytes depends on many factors. The attachment of melanocytes to the basement membrane is possible thanks to proteins such as laminin and collagen IV. Additionally, the secretion of soluble factors by melanocytes, some similar to those secreted by keratinocytes, such as HGF (hepatocyte growth factor) or SCF (stem cell factor) depends on dermal fibroblasts. Moreover, the release of HGF and SCF by fibroblasts depends on the stimulation of IL-1a and TNFα produced by keratinocytes [1]. The melanin that is produced by melanocytes interacts with the keratinocytes to form the functional epidermal melanin unit (EMU). Melanin absorbed and stored in keratinocytes protects DNA against ultraviolet damage, reducing the risk of mutations. This intercommunication between cells is vital to maintain skin homeostasis and to regulate dermal and epidermal cell differentiation and proliferation [10]. One of the popular commercially available 3-D skin models is MelSkin, in which the melanocyte compartment is biologically fully functional [1]. Still, that skin model was created from cells isolated from patients skin biopsies [10,11,12,13].

Here, we describe a simple protocol for preparation of the skin model consisted of commercially available keratinocytes, fibroblasts, and melanocytes cell lines. Because in our 3D model, the keratinocytes are differentiated, this model is closer to the natural skin.

## 2. Results

### 2.1. Scaffold Crosslinking

We investigated the effect of 11 concentrations of genipin on the collagen constructs crosslinking The analysis was repeated 4 times. Collagen contraction was calculated according to the following formula:(1)$$Average\;contraction\;\left[\%\right]=100-\frac{average\;construct\;diameter\;\left[mm\right]}{well\;diameter\;\left[mm\right]}\times 100$$
where well diameter is 7.5 mm.

The addition of 20 µM of genipin reduced contraction of collagen construct from 59% to 24% at the 12th day of construct culture. Lower concentrations of genipin didn’t significantly affect the contraction of collagen. Concentrations in the range from 80 to 200 µM reduced contraction to 14%, on average. These results are presented in Table 1.

### 2.2. Cytotoxicity of Genipin

We have shown the effect of genipin on the keratinocyte (range 1–160 µM) and fibroblast (range 1–400 µM) cell culture. As per ISO 10993-5, the genipin is not considered cytotoxic to the keratinocytes up to the concentration of 10 µM. In the case of fibroblast culture, the genipin is not considered to be cytotoxic up to the concentration of 150 µM. In agreement with the above ISO we did not observe significant effects on keratinocytes and fibroblasts cultured in 10 μM and 150 μM genipin, respectively. The results are presented in Figure 1.

### 2.3. Histologic and Immunostaining Analysis

The haematoxylin and eosin staining demonstrated that the morphology of fibroblasts and keratinocytes in our HSE was similar to that in normal human skin. The analysis showed the collagen hydrogel matrix populated with fibroblasts, and well-defined epidermis on the top (Figure 2A). Immunohistochemical staining demonstrated the presence of vimentin-positive fibroblasts in the dermal layer. Cytokeratin 14 was strongly expressed in the keratinocytes forming the basal layer, and lower expression was observed in the keratinocytes within the more apical layers, stratum spinosum, and stratum granulosum. The cytokeratin 1 expression was detected in the fully differentiated epidermis (Figure 2B).

The addition of the melanocytes to the cell culture did not change the structure of the model. Melanocytes were located in the epidermis and were evenly distributed in the stratum basale. This is a typical distribution of melanocytes in normal skin. Melanocytes were distributed evenly as seen by the even dark color over the entire surface of the construct (Figure 3).

### 2.4. Cytokines Analysis

Homeostasis of the skin is achieved by continuous secretion of chemokines, cytokines, and growth factors by the resident cells. Therefore, we determined the basal secretion profile of GM-CSF (granulocyte-macrophage colony-stimulating factor), IL-10, IL-12p70, IL-15, IL-1α, IL-6, IL-7, IL-8, (interleukins), MCP-1 (monocyte chemoattractant protein-1), TNFα (tumor necrosis factor α), and VEGF (vascular endothelial growth factor) in the 24-h HSE culture supernatant. The analysis showed that longer differentiation of constructs leads to higher secretion of GM-CSF, IL-10, IL-15, IL-1α, IL-6, IL-7, IL-8, and MCP-1. Interestingly, a longer differentiation time leads to the more stable secretion of all analytes, which is reflected by the coefficient of variation. Results of the cytokine and chemokine secretion analysis are presented in Figure 4.

## 3. Discussion

Cell culture technique was discovered over 100 years ago, when Ross Harrison described the method of growing nerve cells, isolated from the frog embryo, in the hanging drop of the medium [14]. Since that time, a revolution in the methods and approach to cell culture has begun, and scientists learned how to isolate and culture cells from different tissues [15]. However, researchers observed that in some cases, the two-dimensional (2D) cell culture does not answer the questions about the structure of the tissue, and cell to cell interactions, and therefore the effect of the agents tested can’t be properly analyzed [16]. The better solution would be the preparation of the 3D constructs but this is difficult. The main problems apply to choosing an appropriate concentration, and timing of the addition of cells to the construct, appropriate media selection, or reproducibility of obtained models [17]. 

In this study, we present a method for the generation of functional full-thickness human skin equivalents which is simple, cost-effective, and reproducible. The main aim of the work was to find simple method for the development of multi-culture HSE from the cells of different origin (commercially available or isolated form the patients) commonly used for the in vitro research. The novel medical devices, drugs, and food additives require biological evaluation but tests on the animals are very restricted by the regulations about the use of animals in research (the so called 3R). Alternative approaches provide opportunities to advance the “Three Rs” and develop a better and more predictive scientific tools. Therefore, in presented study, we did not evaluate the potential of our model for the engraftment treatment of trauma but its use as a simple and repeatable model for the in vitro studies. The first challenge of our model was the stabilization of the collagen scaffold, which in prolonged culture (up to 18th day) show a tendency to contract. Therefore to improve mechanic and enzymatic properties of the scaffold the cross-linked collagen factors should be used [18]. We used genipin, as the cross-link stabilator of the collagen scaffold for the fibroblast culture. Genipin was also used by Meng and Shen [19] for the stabilization of collagen for HSE production. The authors observed that 0.4 mmol/L genipin is non-toxic for the fibroblasts and greatly reduces the contraction of 3D skin equivalents from 87% to 28% (*n* = 9, *p* < 0.05) over a 21-day follow-up period. In our research, we used 20 times lower concentration of genipin, while maintaining appropriate parameters of the collagen scaffold. Interestingly, we also observed lack of the toxic effect of genipin, in range 1–150 µM, on the proliferation of the fibroblast cell line. In contrast, there was a significant reduction of the keratinocytes count as measured by the MTT assay. This suggests that the keratinocytes may be much more sensitive to the genipin than fibroblasts, and therefore the lower concentration of the genipin or longer time after crosslinking of collagen should be considered when seeding the keratinocytes. Another study showed that the high (0.5–10 mM) concentration of genipin affects the proliferation of the human umbilical cord-derived mesenchymal stem cells in the genipin-crosslinked extracellular matrix hydrogel [20].

The next step of our protocol was to establish a simple and cost-effective protocol for the co-culture of keratinocytes and fibroblasts.

Currently, the production of 3D full-thickness skin equivalents requires expensive culture media and a number of different reagents/additives. Such models were described by Reijnders et al. [21], Carlson et al. [22], and Rossi et al. [23]. These models are not only expensive but also labor-intensive. Therefore, we decided to explore the possibility of creating the 3D skin model using ready to use reagents that require as few modifications into media composition as possible. We have successfully generated models using KGM Gold medium and DMEM with Medium 199 combination however we finally decided to proceed with Epilife Medium and Medium 106 combination as it was more cost-efficient and less labor-intensive. We were able to generate models using commercially available primary cells (PCS-200-010, PCS-201-010), immortalized cell lines (CRL-4048, CRL-4001), and primary cells isolated from volunteers (data not shown).

Presented constructs closely resemble the morphology of normal human skin and consists of fibroblast-populated dermal scaffold and well-differentiated epidermis. Described HSEs are generated using commercially available cell lines that are not bound by ethical or logistic issues and can be used at least up to 14–18th passage. The results of histology and immunofluorescence analysis of our constructs were also similar to those obtained by other researchers [24,25]. Many of the HSE described by others were constructed from the cells isolated from patients. However, such patient-derived cells are characterized by a high donor to donor variation. In our method, the unlimited availability of identical cells translates to higher consistency and reproducibility of results. Our model is built from the commercially available cell lines, which reduces time, costs, and does not require very experienced laboratory personnel.

The next step in the production of our 3D type HSE was the addition of melanocytes. The main challenges of the melanocyte’s addition were: 1. choosing the right medium and 2. time of the addition to the constructs. We observed that the best results were obtained when the melanocytes were seeded after mixing with keratinocytes. The full protocol is available in Appendix A as Protocol 1. Histologic analysis of three cell types present in the HSE showed that the addition of melanocytes did not disrupt the arrangement of the keratinocytes and fibroblasts in the scaffold. Moreover, keratinocytes were well-differentiated, which suggested that melanocyte addition did not disrupt cell differentiation processes. Also, the melanocytes were distributed evenly within the whole surface of the construct.

The last step of our protocol was an evaluation of the reproducibility of our HSEs. We observed that the construct organization and cell arrangement, localization, and number were almost identical between different HSEs. Because these observations were burdened by the subjective-judgment error, we chose cytokines evaluation as the more objective and sensitive criterium for analysis of consistency between individual HSEs. We also wanted to know the differences between 2-cell and 3-cell type models in terms of the cytokine expression profiles. Cytokines and chemokines are essential for skin homeostasis, and the skin response to internal and external factors can be measured by changes in its secretome [26]. Here, we showed that the human skin constructs secreted a broad range of chemokines and cytokines consistent with those secreted by native human skin [27,28]. Analysis of the basal secretion levels of HSEs at day 12 and 14 showed higher secretion of GM-CSF, IL-10, IL-15, IL-1α, IL-6, IL-7, IL-8, and MCP-1 at 14 day. This was predictable and may be explained by the ongoing proliferation of cells in the constructs, in which fibroblast fill gaps in the collagen matrix, and keratinocytes form differentiated epidermis. Interestingly, the longer differentiation time leads to a more stable secretion of all analytes indicating that the mature constructs strive to achieve a state of equilibrium. Rejinders et al. [26] showed that in the HSE constructed by TERT-immortalization of human keratinocytes and fibroblasts there was about 55% of the variation in IL-6 (in our model: 5.05%) and 46% in MCP-1 (in our model 6.36%) levels at 14th day of differentiation [21]. Moreover, the authors did not found secretion of vascular endothelial growth factor (VEGF) which was found in normal skin fragmets and in our model. Although the HSE presented in this study has limitations such as the absence of vascularization, lack of mechanical testing, and no in vivo data due to the use of immortalized cell lines, the consistency between the individual HSEs produced in our study indicates that they can be very useful for medical research and testing. Finally, the smaller variability of biological models translates to a smaller sample size needed to achieve statistically significant results [29].

## 4. Material and Methods

### 4.1. Reagents

Cells were purchased from ATCC collection (LGC Standards, Poland), and cell culture reagents from ThermoFisher Scientific, Warsaw, Poland. Cell culture plastics were from BD Falcon (Diag-Med, Warsaw, Poland). Reagents used for cytokine and chemokine secretion analysis were purchased from Merc (Darmstadt, Germany), and antibodies used for immunohistological analysis were from Abcam (Cambridge, UK), LSBios (Seattle, washington, USA), and BioLegend (Koblenz, Germany). Other chemicals were from Sigma Aldrich, Poznan, Poland. Precise protocols are provided as Appendix A.

### 4.2. Cells

PCS-200-010, Primary Epidermal Keratinocytes were cultured in the EpiLife Medium with the addition of Human Keratinocyte Growth Supplement [HKGS], penicillin (100 U/mL), and streptomycin (100 µg/mL). The culture medium was changed three times per week. The cells were cultured up to 14th passage without changes in morphology or ability to generate the HSE.

PCS-201-010, Primary Dermal Fibroblast was cultured in Medium 106 with the addition of Low Serum Growth Supplement (LSGS), penicillin (100 U/mL), and streptomycin (100 µg/mL). The culture medium was changed three times per week. The cells were cultured up to 18th passage without changes in morphology or ability to generate HSE.

PCS-200-013, primary epidermal melanocytes were cultured in Medium 254CF with the addition of human melanocyte growth supplement (HMGS), 200 µM calcium chloride [LSGS], penicillin (100 U/mL), and streptomycin (100 µg/mL). The culture medium was changed three times per week. The cells were cultured up to 6th passage without changes in morphology or ability to generate HSE.

CRL-4048, also known as Ker-CT, Epidermal Keratinocytes immortalized by human telomerase and CDK4, were cultured in KBM Gold Keratinocyte Growth Basal Medium with the addition of KGM Gold Keratinocyte Growth Medium SingleQuots Supplements and Growth Factors, penicillin (100 U/mL), and streptomycin (100 µg/mL).

CRL-4001, also known as BJ-5ta, human fibroblasts immortalized with hTERT, were cultured in 4:1 DMEM and Medium 199 with the addition of 0. 01 mg/mL hygromycin B, 10% fetal bovine serum, penicillin (100 U/mL), and streptomycin (100 µg/mL).

### 4.3. Extracellular Scaffolding

The HSEs were constructed according to Protocol 1 provided in Appendix A. Briefly, rat collagen (3 mg/mL) with 20 µM of genipin was populated with fibroblasts (2.5 × 10^5^/mL) and incubated at 37 ± 1 °C (humidified) with 5 ± 1% CO_2_ for 3 days in Medium 106 with LSGS and antibiotics. Then, keratinocytes (2 × 10^5^/per insert) (mixed with 5–10% of melanocytes or alone) were seeded onto the collagen scaffold containing fibroblasts. The constructs were incubated submerged for 3 days with EpiLife Medium. After 3 days, the inserts with the constructs were cultured in the air-liquid interface in the EpiLife Medium supplemented with HKGS, antibiotics, CaCl_2_, KGF, and ascorbic acid. The culture medium was replaced every two days. The supernatant was collected for the cytokine analysis at the 12th and 14th days of keratinocytes differentiation.

### 4.4. Genipin Crosslinking of Collagen Scaffold

The effect of the addition of genipin (1 µM, 5 µM, 10 µM, 15 µM, 20 µM, 30 µM, 50 µM, 80 µM, 100 µM, 150 µM, and 200 µM) on the collagen contraction in HSE was investigated by measuring the diameter of collagen constructs at the 12th day of culture.

### 4.5. Cytotoxicity Analysis

The genipin cytotoxic effect on the fibroblasts (CRL-4001) and keratinocytes (CRL-4048) were analyzed using an MTT assay (Sigma Aldrich, Poznan, Poland) according to the modified ISO 10993-5 method. Briefly, octuplicate monolayers of appropriate (fibroblasts or keratinocytes) cells were dosed with various concentrations of genipin in medium and incubated at 37 ± 1 °C in the presence of 5 ± 0.1 % CO_2_ for 24 ± 1 h. We used the full range of genipin concentration (from 1 to 400 µM). However, to better show its cytotoxic effect, we presented in the figure only these doses which were on the border of cytotoxicity. Therefore, in Figure 1 the range of genipin in the keratinocyte culture is 0–160 µM, and 0–400 µM in the fibroblast culture. Following the incubation, the MTT solution, prepared just before use, was added into each well. After 120 ± 15 min incubation at 37 ± 1 °C (humidified) in 5 ± 0.1 % CO_2_, the MTT solution was replaced with isopropanol, and cultures were incubated for 10 min at 37 ± 1 °C (humidified) in the presence of 5 ± 0.1 % CO_2_. The percentage of the viable cells in each sample was determined by a comparison to the blanks. A decrease in the number of live cells resulted in a decrease of metabolic activity in the sample. This decrease directly correlates with the amount of blue-violet formazan formed, as monitored by the optical density at 570 nm with a differential filter of 650 nm were measured by a Fluostar Omega microplate reader (FLUOstar Omega, BMG Labtech, Ortenberg, Germany).

### 4.6. Histological Analysis

For the histological analysis, the HSEs from at least five independent experiments were fixed in 4% formaldehyde and embedded in paraffin. Thereby, 5 µm paraffin sections were placed on the microscope slides (Protocol 2) and used for the haematoxylin and eosin (Protocol 3) and immunohistochemistry (Protocol 4). For the HE stains the Gill II hematoxylin and Shandon Alcoholic Eosin Y (ThermoFisher) were used. Brightfield photographs were captured using Zeiss AXIO (200X). For IHC staining, citrate buffer was used for antigen retrieval. Following antibodies were used:anti-cytokeratin 1 antibody, clone LHK1, DY550, REF#: LS-C180221, LSBioanti-cytokeratin 14 antibody, clone LL002, FITC, REF#: ab77684, abcamanti-vimentin antibody clone O91D3, Alexa Fluor^®^ 647, REF#: 677807, BioLegend

Fluorescence photographs were made using Leica TCS SP5 confocal microscope and LAS X software (200X).

### 4.7. Cytokine Analysis

Keratinocytes and fibroblasts have been reported to secrete a plethora of cytokines and chemokines including but not limited to IL-1, IL-6, IL-7, IL-8, IL-10, IL-12, IL-15, TNFα, CSF, VEGF, and MCP-1 [30,31,32,33]. Therefore, to evaluate consistency between the constructs we decided to measure the concentration of cytokines and chemokines listed below. The concentration of cytokines, chemokines, and growth factors in the 24h supernatant samples collected from HSEs was determined using the Luminex MagPix platform. The HCYTOMAG-60K-11 Kit (Merck, Poland) was used for the analysis of GM-CSF, IL-10, IL-12p70, IL-15, IL-1α, IL-6, IL-7, IL-8, MCP-1, TNFα, and VEGF. The basal secretion profiles of HSEs at the 12th and 14th day after initiation of keratinocytes differentiation were analyzed. The analyses were performed according to the manufacturer’s specifications. Results from 6 independent experiments (double analysis of the same sample) are presented as a mean ± SEM pg/mL.

### 4.8. Statistical Analysis

Statistical evaluation of the results was performed using T-tests and one-way ANOVA with Bonferroni correction (in the case of a normal distribution) or non-parametric Kruskal–Wallis and Mann–Whitney U tests (in the case of an abnormal distribution). Assessment of the distribution of the data was evaluated using the Shapiro–Wilk test. GraphPad Prism software was used to carry out these tests (version 7; GraphPad Software, Inc., La Jolla, CA, USA). *p* < 0.05 was considered a statistically significant difference.

## 5. Conclusions

In conclusion, in this manuscript, we described simple, reliable, and cost-effective full-thickness human skin equivalents that can be used in scientific and industrial research without the logistic and ethical limitations associated with traditional HSEs. Moreover, the determined basal cytokine and chemokine secretion profile and low variability makes the described HSEs a potentially important tool that could be used in drug delivery, cytotoxicity, irritancy, and sensitization studies. Finally, the use of alternative research models, such as this one, supports the global policy of reducing animal research.

## Figures and Tables

**Figure 1 ijms-21-04644-f001:**
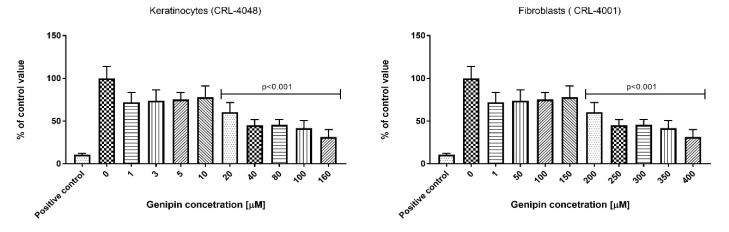
Effect of genipin on keratinocytes and fibroblasts. The cytotoxic effect was measured by MTT test according to ISO 10993-5 protocol.

**Figure 2 ijms-21-04644-f002:**
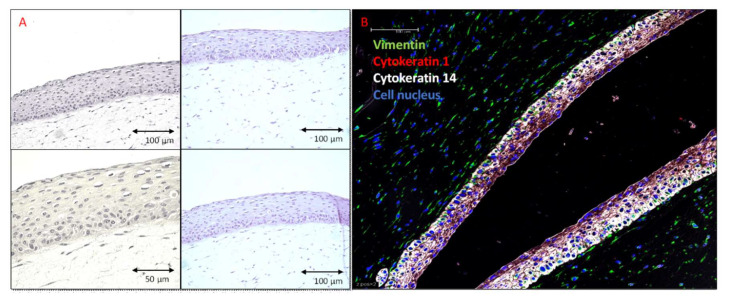
Human skin equivalent generated using PCS-200-010 and PCS-201-010 cell lines and differentiated for 14 days. Sections of the formalin-fixed, and paraffin-embedded, 3-dimensional (3D) skin models. (**A**) Hematoxylin and Eosin staining. HSEs consist of well-differentiated epidermis on top of a fibroblast-populated dermis. (**B**) Fluorescent immunostaining. Green—Vimentin (Biolegend 677807), Red—Cytokeratin 1 (LSBio LS-C180221), White—Cytokeratin 14 (Abcam ab77684), Blue—Cell nucleus (DAPI). Cytokeratin 14 is strongly expressed in keratinocytes forming a basal layer with lower expression in the keratinocytes of the more apical layers, stratum spinosum, and stratum granulosum. Cytokeratin 1 expression is detected in the fully differentiated epidermis.

**Figure 3 ijms-21-04644-f003:**
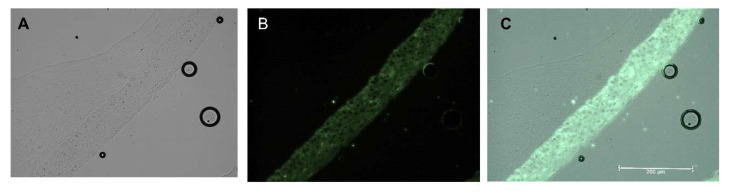
Human skin equivalent generated using PCS-200-010, PCS-200-013, and PCS-201-010 cell lines and differentiated for 14 days. Formalin-fixed, paraffin-embedded 3-dimensional (3D) skin models. Fluorescent immunostaining. Green—Cytokeratin 14 (Abcam ab77684). Melanocytes are seen as the dark color of the entire surface of the epidermis (green) layer. (**A**) brightfield, (**B**) Cytokeratin 14 (green), (**C**) merge.

**Figure 4 ijms-21-04644-f004:**
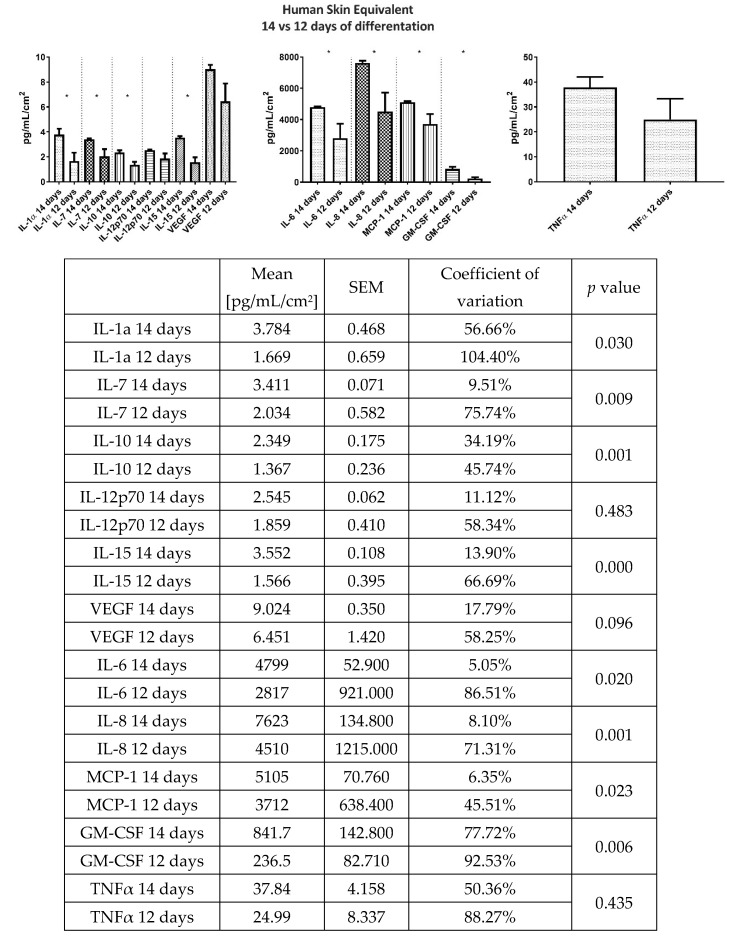
Comparison of GM-CSF (granulocyte-macrophage colony-stimulating factor), IL-10, IL-12p70, IL-15, IL-1α, IL-6, IL-7, IL-8, (interleukins), MCP-1 (monocyte chemoattractant protein-1), TNFα (tumor necrosis factor α), and VEGF (vascular endothelial growth factor) levels in 24 h culture supernatant between 12 and 14 days differentiated HSEs. Results presented as Mean ± SEM [pg/mL/cm^2^]. The variability between the samples is measured by the coefficient of variation.

**Table 1 ijms-21-04644-t001:** The effects of genipin addition on the collagen contraction (*n* = 4).

Condition.	Genipin Concentration [µM]	Average Construct Diameter [mm]	Average Contraction [%]
1	0	3.1	59
2	1	3.4	55
3	5	3.8	49
4	10	4.3	43
5	15	4.7	37
6	20	5.7	24
7	30	5.9	21
8	50	6.1	19
9	80	6.4	15
10	100	6.5	13
11	150	6.4	15
12	200	6.6	12

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
