# Peer review of "A Simple Method for the Production of Human Skin Equivalent in 3D, Multi-Cell Culture"

_ijms, 2020, doi:10.3390/ijms21134644_

Round 1

Reviewer 1 Report

Manuscript ID: ijms-836415

The manuscript entitled 'Simply and repeatable method for production of human skin equivalent in 3D, multi-cells culture in vitro' by Sławomir Lewicki’s group proposed a potential useful, cost-effective, and easy method for preparation of HSE. The results presented are simple and well-understandable. This kind of research papers are valuable for researchers in related research fields. However, modifications are necessary before proceeding publication process although the reviewer thinks this MS is interesting. The points are listed below.

  1. I am not a native English speaker, but I feel the English text in this MS must be carefully checked again by a native speaker. For example, in abstract and introduction,

Lines 27-28: I cannot understand the sentence, ‘Genipin is not considered to be cytotoxic to keratinocytes up to a concentration of 10 μM and 250 μM for fibroblast.’

Lines 47: ‘appendages as, for example dermal…’ à ‘appendages, for example, dermal…’

Line 65: What is more ‘,’

Line 67: other hand ‘,’

Please check whole MS again.

  1. Line 25: ’20 mM’ is ’20 μM’?

  1. Line 53: ‘3-dimensional ‘(3-D)’’. (Please show an abbreviation)

  1. Lines 177-179: Genipin is not cytotoxic under 10 μM and 250 μM to keratinocytes and fibroblasts, respectively? The reviewer feels effective to a certain degree. In figure 1, only p<0.001 is shown, but you mentioned p<0.05 is significant difference (lines 166-167). Significant differences were not observed under 10 μM and 250 μM? If so, please mention it.

  1. Figure 3: Please identify each panel with using A, B, and C. And mention in the legend.

  1. Line 216: ‘Fig 2’ is ‘Fig 4’?

  1. Line 262: ‘These models are not only cost-effective’: This part is strange. In lines 259-261, you explained about usual expensive methods for HSE, didn’t you?

Author Response

Response to the Reviews

All changes, pertaining to the reviewrs’ suggestions are marked in the track system and the revised manuscript. The responses to all reviewers’ comments are below.  We added 2 manuscripts – with changes (a) and  clear corrected paper (b).

First round  

Review 1:

I am not a native English speaker, but I feel the English text in this MS must be carefully checked again by a native speaker. For example, in abstract and introduction,

Response: The new version of the text was English edited by the native Endglish speaker

Lines 27-28: I cannot understand the sentence, ‘Genipin is not considered to be cytotoxic to keratinocytes up to a concentration of 10 μM and 250 μM for fibroblast.’

Response: The sentence was changed: ‘In agreement with the above ISO we did not observe significant effects on keratinocytes and fibroblasts cultured in 10 μM and 150 μM genipin, respectively.

 Lines 47: ‘appendages as, for example dermal…’ à ‘appendages, for example, dermal…’

Line 65: What is more ‘,’

Line 67: other hand ‘,’

Please check whole MS again.

Response: The whole MS was edited by the native English speaker. The word “appendeges” had been changed.

 Line 25: ’20 mM’ is ’20 μM’?

Response: The typo was corrected.

  Line 53: ‘3-dimensional ‘(3-D)’’. (Please show an abbreviation)

Response: Abbreviation was added.

 Lines 177-179: Genipin is not cytotoxic under 10 μM and 250 μM to keratinocytes and fibroblasts, respectively? The reviewer feels effective to a certain degree. In figure 1, only p<0.001 is shown, but you mentioned p<0.05 is significant difference (lines 166-167). Significant differences were not observed under 10 μM and 250 μM? If so, please mention it.

Response: As was shown in the figure 1 – genipin in concentration 150 μM and below is not cytotoxic for the fibroblasts. The p value <0.05 was considered as stastically significant, and none of the changes shown in the  Figure 1 had  p< 0.05. To make it clearer we added the following sentence to the main text: In agreement with the above ISO we did not observe significant effects on keratinocytes and fibroblasts cultured in 10 μM and 150 μM genipin, respectively

 Figure 3: Please identify each panel with using A, B, and C. And mention in the legend.

Response: We added lettering to the figure.

 Line 216: ‘Fig 2’ is ‘Fig 4’?

Response: This error was corrected.

 Line 262: ‘These models are not only cost-effective’: This part is strange. In lines 259-261, you explained about usual expensive methods for HSE, didn’t you?

Response: we changed the sentence to: These models are not only expensive but also a labor intensive.

Reviewer 2 Report

Manuscript by Szymanski et al. is a concise methodological report describing  descriptive in vitro study of tissue-engineered human skin generated by self-assembly from available cell lines.

In current state the manuscript requires significant improvement due to both, scientific reasons and quality of written presentation.

Below I shall provide crucial points that may require improvement and suggest changes to the manuscript title.

General comments:

1) Title contains a grammatical error using "Simply" where "Simple" is more appropriate

2) Authors stress "repeatable" while very little data is provided to support this points. Is it really necessary to stress this feature?

3) using "culture" and "in vitro" adjacently is a semantic tautology, so I suggest to omit one of descriptive terms used

4) Quality and style of writing is definitely below standards and the text contains numerous stylistical and grammatical errors. I suggest rolling the revised manuscript through a professional editing service as far improvement will require very serious changes in the text.

Major points:

1) Choice of cytokines studied is not supported by a clear physiological rationale

2) Section 3.1 has no illustration or numeric material which might be added as a figure or table 

3) Lines 184-185 present a claim that is not correct - normal skin contains its appendages, fat, blood vessels etc. While the layer structure of HSE mimics human skin it has limited ability to reproduce native organ

4) Figure 4 table is barely readable due to small font

5) Authors claim differentiation in Figure 2 legend while it is not clear why this term is applied - no progenitor or precursor cells were added to create HSE - thus, hardly there was a shift in cell maturity or phenotype besides a vivid self-organisation of HSE in vitro

6) Discussion has approx 1/4 of text describing general problems and history of cell biology while not giving insights into achievements of the work.

7) Study has numerous limitations including lack of instrumental mechanical properties assessment, in vivo test to evaluate engraftment, neglected aspects of angiogenesis and long-term cell fate.

8) The study has tissue mediocre originality with a focus on methodological approach and descriptive results rather that investigation into mechanisms of phenomena observed

9) use of commercial cell lines marginally excludes therapeutic application of established protocol thus reducing the potential of protocol to in vitro modelling of human skin. Thus, to create a therapeutic product one will require to use a primary cell source from auto- or allogeneic skin and this creates a conflict with claimed study rationale (lines 43 and further) to avoid cell isolation.

Minor points:

1) Line 18 - HSE in not correctly abbreviated while correct abbraviation and its wording appears only in line 91

2) Line 47 skin and skin appear adjacently creating a tautology

3) Line 68 "Melanin absorber...." - the sentence fails to deliver a reasonable message due to grammar inconsistency

4) Line 122 - "Cytotoxic" used while "Cytotoxicity" is more appropriate

Author Response

Response to the Reviews

All changes, pertaining to the reviewrs’ suggestions are marked in the track system and the revised manuscript. The responses to all reviewers’ comments are below.  We added 2 manuscirpts – with changes (a) and  clear corrected paper (b).

Reviewer 2

Manuscript by Szymanski et al. is a concise methodological report describing descriptive in vitro study of tissue-engineered human skin generated by self-assembly from available cell lines. In current state the manuscript requires significant improvement due to both, scientific reasons and quality of written presentation. Below I shall provide crucial points that may require improvement and suggest changes to the manuscript title.

General comments:

Response:

1) Title contains a grammatical error using "Simply" where "Simple" is more appropriate

Response: We changed the title to:   A simple method for the production of human skin equivalent in 3D, multi-cell culture.

2) Authors stress "repeatable" while very little data is provided to support this points. Is it really necessary to stress this feature?

Response: We changed the text accordingly. We removed the word “repeatable” from parts of the text.

3) using "culture" and "in vitro" adjacently is a semantic tautology, so I suggest to omit one of descriptive terms used

Response: we changed the title to: A simple method for the production of human skin equivalent in 3D, multi-cell culture.

4) Quality and style of writing is definitely below standards and the text contains numerous stylistical and grammatical errors. I suggest rolling the revised manuscript through a professional editing service as far improvement will require very serious changes in the text.

Response: The manuscript was edited by the native English speaker.

Major points:

1) Choice of cytokines studied is not supported by a clear physiological rationale

Response: : Assessment of cytokine secretion in the multi-culture population is a difficult challenge. In our model there are 3 different cell lines: the epithelial (keratinocytes), mesenchymal (fibroblasts), and neural crest cells (melanocytes). For such complicated model it is hard to find ideal physiological rationale for choosing the particular cytokine secretion profile. Moreover different type of cells from the same germ layer i.e. epithelial will have different profile of cytokines secretion. Besides, the specific relationships between the cells in the multi-culture model will affect the cell secretion profiles, and/or the concertation of particular cytokines. Previously, in the study by Szymanski et al. (Szymanski L, Cios A, Lewicki S, Szymanski P, Stankiewicz W. Fas/FasL pathway and cytokines in keratinocytes in atopic dermatitis - Manipulation by the electromagnetic field. PLoS One. 2018;13(10):e0205103. Published 2018 Oct 4. doi:10.1371/journal.pone.0205103) we found that the secretion profiles of cytokines in culture of  keratinocytes isolated from patients varied between individuals . Therefore, in present work to assessed the consistency of our muli-cell models we assessed their cytokines secretion profiles. As stated in the point 2.7 “Keratinocytes and fibroblasts have been reported to secrete a plethora of cytokines and chemokines including but not limited to IL-1, IL-6, IL-7, IL-8, IL-10, IL-12, IL-15, TNFα, CSF, VEGF, and MCP-1 [14–17]”. Based on the literature we have chosen “ready to use panel” from Merck (HCYTOMAG-60K-11 Kit) that contains as many cytokines produced by keratinocytes, melanocytes and fibroblast as possible. We decided to use ready to use panel instead of building our own as this approach was more cost effective.

2) Section 3.1 has no illustration or numeric material which might be added as a figure or table

Response: We added Table 1. The effects of genipin addition on the collagen contraction (n=4). and edited the  relevant text.

3) Lines 184-185 present a claim that is not correct - normal skin contains its appendages, fat, blood vessels etc. While the layer structure of HSE mimics human skin it has limited ability to reproduce native organ

Response: We changed the sentence to: The Haematoxylin & Eosin staining demonstrates that the morphology of the fibroblasts and keratinocytes was similar (identical) to that of the native human skin.

4) Figure 4 table is barely readable due to small font

Response: The figure was changed to be more readeable.

5) Authors claim differentiation in Figure 2 legend while it is not clear why this term is applied - no progenitor or precursor cells were added to create HSE - thus, hardly there was a shift in cell maturity or phenotype besides a vivid self-organisation of HSE in vitro

Response: To produce our model we used keratinocytes from the commercially available primary cells lines. These cells were isolated form the normal skin. The cells exhibited normal proliferation potential, which strongly suggested that the cells remained undifferentiated, and had a potential to differentiate under appropriate conditions. These conditions were well described, and characterized in the liquid-air culture (Pruniéras M, Régnier M, Woodley D. Methods for cultivation of keratinocytes with an air-liquid interface. J Invest Dermatol. 1983;81(1 Suppl):28s-33s. doi:10.1111/1523-1747.ep12540324). Moreover, in normal skin, the differentiation of keratinocytes is assessed by the changes in their morphology, and the changes in the expression of cell-surface markers: the cytokeratin 1 (keratinocytes from terminally differentiating epidermis), and cytokeratin 14 (keratinocytes from the basal layer). These changes were also included in our model (Fig. 2).

6) Discussion has approx 1/4 of text describing general problems and history of cell biology while not giving insights into achievements of the work.

Response: We shortened the part describing the general problems and history.

7) Study has numerous limitations including lack of instrumental mechanical properties assessment, in vivo test to evaluate engraftment, neglected aspects of angiogenesis and long-term cell fate.

Response: The main aim of the work was to find simple method for the development of multi-culture HSE from the cells of different origin (commercially available or isolated form the patients) commonly used for the in vitro research. The novel medical devices, drugs, and food additives require biological evaluation but the tests on the animals are very restricted by the regulations about the use of animals in research (the so called 3R). Alternative approaches provide opportunities to advance the "Three Rs" and develop a better and more predictive scientific tools. Therefore, in presented here study we did not evaluate the potential of our model for the engraftment treatment of trauma but its use as a simple and repeatable model for the  in vitro studies. We belive that the model has a high potential for the use in the skin engraftments and we will explore this option in the future studies.

8) The study has tissue mediocre originality with a focus on methodological approach and descriptive results rather that investigation into mechanisms of phenomena observed

Response: We agree with the Reviewer 2. Explanations of our proof of concept were described in sections 5) and 7) in “Response to the Reviewer 2”.

9) use of commercial cell lines marginally excludes therapeutic application of established protocol thus reducing the potential of protocol to in vitro modelling of human skin. Thus, to create a therapeutic product one will require to use a primary cell source from auto- or allogeneic skin and this creates a conflict with claimed study rationale (lines 43 and further) to avoid cell isolation.

Response: We agree with the Reviewer 2. As we mentioned above – we wanted to find a good protocol for the development of the  multi-culture (3 cell types) and differentiated HSE, which could be used as an alternative methods to the animals. We belive that the engrafting of the model formed from the cells with HLA proteins to live organism will create the immune response and rejection of the engraft. However, as mentioned in point 1) of  major points, our previous study showed high differences in the  level of secreted cytokines from humans. Cytokines concertation is one of the measurable effect of irritation etc. in: Biological evaluation of medical devices (ISO 10993-10:2010, Biological evaluation of medical devices — Part 10: Tests for irritation and skin sensitization https://www.iso.org/standard/40884.html).

Minor points:

1) Line 18 - HSE in not correctly abbreviated while correct abbreviation and its wording appears only in line 91

Response: HSE is now correctly abbreviated in the abstract.

2) Line 47 skin and skin appear adjacently creating a tautology

Response: We changed the wording.

3) Line 68 "Melanin absorber...." - the sentence fails to deliver a reasonable message due to grammar inconsistency

Response: The sentence was changed to: Melanin absorbed and stored in keratinocytes protects DNA against ultraviolet damage reducing the risk of mutations

4) Line 122 - "Cytotoxic" used while "Cytotoxicity" is more appropriate

Response: The point 2.5 was changed accordingly.

Round 2

Reviewer 2 Report

Overall, most queries were responded properly.

However, I suggest to add study limitations as a clearly outlined part of Discussion to avoid Reader's confusion.

Limitations were listed in my Round 1 review including commercial immortalised line usage, lack of mechanical testing, absence of vacsularisation and absence of in vivo data. Authors do state that the model is mainly for in vitro use and substance testing, so I suggest to alter the text to make this scope of application and study reported visible to the Reader.

Best regards, PM

Author Response

Reviewer 2

However, I suggest to add study limitations as a clearly outlined part of Discussion to avoid Reader's confusion.

Limitations were listed in my Round 1 review including commercial immortalised line usage, lack of mechanical testing, absence of vacsularisation and absence of in vivo data. Authors do state that the model is mainly for in vitro use and substance testing, so I suggest to alter the text to make this scope of application and study reported visible to the Reader.

Response: We added in dissusion the sensteces:

  1. at the beginingn of dissusion:

“The main aim of the work was to find simple method for the development of multi-culture HSE from the cells of different origin (commercially available or isolated form the patients) commonly used for the in vitro research. The novel medical devices, drugs, and food additives require biological evaluation but the tests on the animals are very restricted by the regulations about the use of animals in research (the so called 3R). Alternative approaches provide opportunities to advance the "Three Rs" and develop a better and more predictive scientific tools. Therefore, in presented study we did not evaluate the potential of our model for the engraftment treatment of trauma but its use as a simple and repeatable model for the in vitro studies.”

  1. At the end of dissucusion:

“Although, the HSE presented in this study has limitations such as the absence of vascularization, lack of mechanical testing, and no in vivo data due to the use of immortalized cell lines the consistency between the individual HSEs produced in our study indicates that they can be very useful for medical research and testing”.